# Causal Inference and Mechanism Clustering of A Mixture of Additive Noise Models

**Shoubo Hu**[*], **Zhitang Chen**[†], **Vahid Partovi Nia**[†], **Laiwan Chan**[*], **Yanhui Geng**[‡]

[*]The Chinese University of Hong Kong; [†]Huawei Noah's Ark Lab;
[‡]Huawei Montréal Research Center
[*]{sbhu, lwchan}@cse.cuhk.edu.hk
[†‡]{chenzhitang2, vahid.partovinia, geng.yanhui}@huawei.com

## Abstract

The inference of the causal relationship between a pair of observed variables is a fundamental problem in science, and most existing approaches are based on one single causal model. In practice, however, observations are often collected from multiple sources with heterogeneous causal models due to certain uncontrollable factors, which renders causal analysis results obtained by a single model skeptical. In this paper, we generalize the Additive Noise Model (ANM) to a mixture model, which consists of a finite number of ANMs, and provide the condition of its causal identifiability. To conduct model estimation, we propose Gaussian Process Partially Observable Model (GPPOM), and incorporate independence enforcement into it to learn latent parameter associated with each observation. Causal inference and clustering according to the underlying generating mechanisms of the mixture model are addressed in this work. Experiments on synthetic and real data demonstrate the effectiveness of our proposed approach.

## 1   Introduction

Understanding the data-generating mechanism (g.m.) has been a main theme of causal inference. To infer the causal direction between two random variables (r.v.s) $X$ and $Y$ using passive observations, most existing approaches first model the relation between them using a functional model with certain assumptions [18, 6, 21, 8]. Then a certain asymmetric property (usually termed *cause-effect asymmetry*), which only holds in the causal direction, is derived to conduct inference. For example, the additive noise model (ANM) [6] represents the effect as a function of the cause with an additive independent noise: $Y = f(X) + \epsilon$. It is shown in [6] that there is no model of the form $X = g(Y) + \tilde{\epsilon}$ that admits an ANM in the anticausal direction for most combinations $(f, p(X), p(\epsilon))$.

Similar to ANM, most causal inference approaches based on functional models, such as LiNGAM [18], PNL [21], and IGCI [9], assume a single causal model for all observations. However, there is no such a guarantee in practice, and it could be very common that the observations are generated by a mixture of causal models due to different data sources or data collection under different conditions, rendering existing single-causal-model based approaches inapplicable in many problems (e.g. Fig. 1). Recently, an approach was proposed for inferring the causal direction of mixtures of ANMs with discrete variables [12]. However, the inference of such mixture models with continuous variables remains a challenging problem and is not yet well studied.

Another question regarding mixture models addressed in this paper is how one could reveal causal knowledge in clustering tasks. Specifically, we aim at finding clusters consistent with the causal g.m.s of a mixture model, which is usually vital in the preliminary phase of many research. For example in the analysis of air data (see section 4.2 for detail), discovering knowledge from air data combined from several different regions (i.e. mechanisms in causal perspective) is much more difficult than

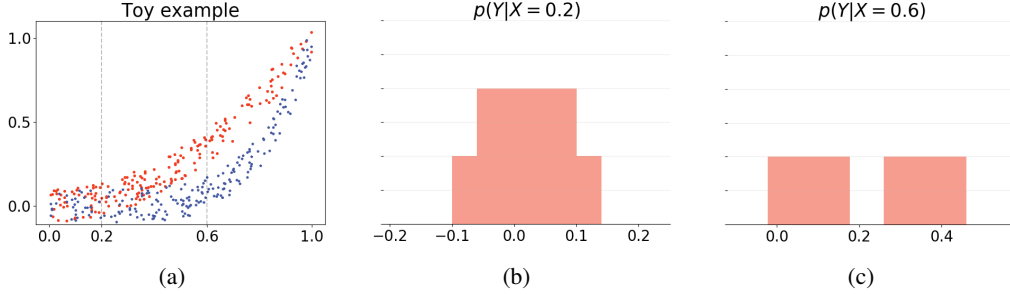

Figure 1: Example illustrating the failure of ANM on the inference of a mixture of ANMs (a) the distribution of data generated from $M_1 : Y = X^2 + \epsilon$ (red) and $M_2 : Y = X^5 + \epsilon$ (blue), where $X \sim U(0,1)$ ($x$-axis) and $\epsilon \sim U(-0.1, 0.1)$ ; (b) Conditional $p(Y|X = 0.2)$; (c) Conditional $p(Y|X = 0.6)$. It is obvious that when the data is generated from a mixture of ANMs, the consistency of conditionals is likely to be violated which leads to the failure of ANM.

from data of each region separately. Most existing clustering algorithms are weak for this perspective as they typically define similarity between observations in the form of distances in some spaces or manifolds. Most of them neglect the relation among r.v.s within a feature vector (observation), and only use those feature dimensions to calculate an overall distance metric as the clustering criterion.

In this paper, we focus on analyzing observations generated by a mixture of ANMs of two r.v.s and try to answer two questions: 1) *causal inference:* how can we infer the causal direction between the two r.v.s? 2) *mechanism clustering:* how can we cluster the observations generated from the same g.m. together? To answer these questions, first as the main result of this paper, we show that the causal direction of the mixture of ANMs is identifiable in most cases, and we propose a variant of GP-LVM [10] named Gaussian Process Partially Observable Model (GPPOM) for model estimation, based on which we further develop the algorithms for causal inference and mechanism clustering.

The rest of the paper is organized as follows: in section 2, we formalize the model, show its identifiability and elaborate mechanism clustering; in section 3, model estimation method is proposed; we present experiments on synthetic and real world data in section 4 and conclude in section 5.

## 2   ANM Mixture Model

### 2.1   Model definition

Each observation is assumed to be generated from an ANM and the entire data set is generated by a finite number of related ANMs. They are called the ANM Mixture Model (ANM-MM) and formally defined as:

**Definition 1** (ANM Mixture Model). *An ANM Mixture Model is a set of causal models of the same causal direction between two continuous r.v.s $X$ and $Y$. All causal models share the same form given by the following ANM:*

$$Y = f(X; \theta) + \epsilon, \qquad (1)$$

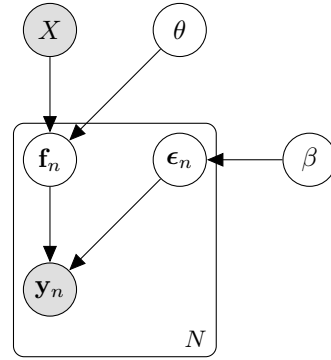

Figure 2: ANM Mixture Model

*where $X$ denotes the cause, $Y$ denotes the effect, $f$ is a nonlinear function parameterized by $\theta$ and the noise $\epsilon \perp\!\!\!\perp X$. The differences between causal models in an ANM-MM stem only from different values of function parameter $\theta$. In ANM-MM, $\theta$ is assumed to be drawn from a discrete distribution on a finite set $\Theta = \{\theta_1, \cdots, \theta_C\}$, i.e. $\theta \sim p_\theta(\theta) = \sum_{c=1}^{C} a_c \mathbf{1}_{\theta_c}(\cdot)$, where $a_c > 0$, $\sum_{c=1}^{C} a_c = 1$ and $\mathbf{1}_{\theta_c}(\cdot)$ is the indicator function of a single value $\theta_c$.*

Obviously in ANM-MM, all observations are generated by a set of g.m.s, which share the same function form ($f$) but differ in parameter values ($\theta$). This model is inspired by commonly encountered cases where the data-generating process is slightly different in each independent trial due to the influence of certain external factors that one can hardly control. In addition, these factors are usually

believed to be independent of the observed variables. The data-generating process of ANM-MM can be represented by a directed graph in Fig. 2.

## 2.2 Causal inference: identifiability of ANM-MM

Let $X$ be the cause and $Y$ be the effect $(X \to Y)$ without loss of generality. As most recently proposed causal inference approaches, following postulate, which was originally proposed in [1], is adopted in the analysis of ANM-MM.

**Postulate 1** (Independence of input and function). *If $X \to Y$, the distribution of $X$ and the function $f$ mapping $X$ to $Y$ are independent since they correspond to independent mechanisms of nature.*

In a general perspective, postulate 1 essentially claims the independence between the cause $(X)$ and mechanism mapping the cause to effect [9]. In ANM-MM, we interpret the independence between the cause and mechanism in an intuitive way: $\theta$, as the function parameter, captures all variability of mechanisms $f$ so it should be independent of the cause $X$ according to postulate 1. Based on the independence between $X$ and $\theta$, cause-effect asymmetry could be derived to infer the causal direction.

Since ANM-MM consists of a set of ANMs, the identifiability result of ANM-MM can be a simple corollary of that in [6] when the number of ANMs $(C)$ is equal and there is a one-to-one correspondence between mechanisms in the forward and backward ANM-MM. In this case the condition of ANM-MM being unidentifiable is to fulfill $C$ ordinary differential equations given in [6] simultaneously which can hardly happen in a generic case. However, $C$ in ANM-MM in both directions may not necessarily be equal and there may also exist many-to-one correspondence between ANMs in both directions. In this case, the identifiability result can not be derived as a simple corollary of [6]. To analyze the identifiability result of ANM-MM, we first derive lemma 1 to find the condition of existence of many-to-one correspondence (which is a generalization of the condition given in [6]), then conclude the identifiability result of ANM-MM (theorem 1) based on the condition in lemma 1. The condition that there exists one backward ANM for a forward ANM-MM is:

**Lemma 1.** *Let $X \to Y$ and they follow an ANM-MM. If there exists a backward ANM in the anti-causal direction, i.e.*

$$X = g(Y) + \tilde{\epsilon},$$

*the cause distribution ($p_X$), the noise distribution ($p_\epsilon$), the nonlinear function ($f$) and its parameter distribution ($p_\theta$) should jointly fulfill the following ordinary differential equation (ODE)*

$$\xi''' - \frac{G(X,Y)}{H(X,Y)}\xi'' = \frac{G(X,Y)V(X,Y)}{U(X,Y)} - H(X,Y), \tag{2}$$

*where $\xi := \log p_X$, and the definitions of $G(X,Y)$, $H(X,Y)$, $V(X,Y)$ and $U(X,Y)$ are provided in supplementary due to the page limitation.*

*Sketch of proof.* Since $X$ and $Y$ follow an ANM-MM, their joint density is factorized in the causal direction by $p(X,Y) = \sum_{c=1}^{C} p(Y|X,\theta_c)p_X(X)p_\theta(\theta_c) = p_X(X)\sum_{c=1}^{C} a_c p_\epsilon(Y - f(X;\theta_c))$. If there exists a backward ANM in the anti-causal direction, i.e. $X = g(Y)+\tilde{\epsilon}$, then $p(X,Y) = p_{\tilde{\epsilon}}(X - g(Y))p_Y(Y)$ and $\frac{\partial}{\partial X}\left(\frac{\partial^2 \pi/\partial X \partial Y}{\partial^2 \pi/\partial X^2}\right) = 0$ holds, where $\pi = \log\left[p_{\tilde{\epsilon}}(X - g(Y))p_Y(Y)\right]$, in the backward ANM. Since $p(X,Y)$ should be the same, by substituting $p(X,Y) = p_X(X)\sum_{c=1}^{C} a_c p_\epsilon(Y - f(X;\theta_c))$ into $\frac{\partial}{\partial X}\left(\frac{\partial^2 \pi/\partial X \partial Y}{\partial^2 \pi/\partial X^2}\right) = 0$, the condition shown in (2) is obtained.

The proof of lemma 1 follows the idea of the identifiability of ANM in [6] and is provided in the supplementary. Since the condition that one backward ANM exists for an forward ANM-MM (mixture of ANMs) is more restrictive than that for a single forward ANM, which is the identifiability in [6], lemma 1 indicates that a backward ANM is unlikely to exist in the anticausal direction if 1) $X$ and $Y$ follow an ANM-MM; 2) postulate 1 holds. Based on lemma 1, it is reasonable to hypothesize that a stronger result, which is justified in theorem 1, is valid, i.e. if the g.m. follows an ANM-MM, then it is almost impossible to have a backward ANM-MM in the anticausal direction.

**Theorem 1.** *Let $X \to Y$ and they follow an ANM-MM. If there exists a backward ANM-MM,*

$$X = g(Y;\omega) + \tilde{\epsilon},$$

*where $\omega \sim p_\omega(\omega) = \sum_{\tilde{c}=1}^{\tilde{C}} b_{\tilde{c}} \mathbf{1}_{\omega_{\tilde{c}}}(\cdot)$, $b_{\tilde{c}} > 0$, $\sum_{\tilde{c}=1}^{\tilde{C}} b_{\tilde{c}} = 1$ and $\tilde{\epsilon} \perp\!\!\!\perp Y$, in the anticausal direction, then $(p_X, p_\epsilon, f, p_\theta)$ should fulfill $\tilde{C}$ ordinary differential equations similar to (2), i.e.,*

$$\xi''' - \frac{G^{(\tilde{c})}(X,Y)}{H^{(\tilde{c})}(X,Y)} \xi'' = \frac{G^{(\tilde{c})}(X,Y) V^{(\tilde{c})}(X,Y)}{U^{(\tilde{c})}(X,Y)} - H^{(\tilde{c})}(X,Y), \quad \tilde{c} = 1, 2, \cdots, \tilde{C}, \tag{3}$$

*where $\xi := \log p_X$, $G^{(\tilde{c})}(X,Y)$, $H^{(\tilde{c})}(X,Y)$, $U^{(\tilde{c})}(X,Y)$ and $V^{(\tilde{c})}(X,Y)$ are defined similarly to those in lemma 1.*

*Proof.* Assume that there exists ANM-MM in both directions. Then there exists a non overlapping partition of the entire data $\mathcal{D} := \{(x_n, y_n)\}_{n=1}^N = \mathcal{D}_1 \cup \cdots \cup \mathcal{D}_{\tilde{C}}$ such that in each data block $\mathcal{D}_{\tilde{c}}$, there is an ANM-MM in the causal direction $Y = f(X;\theta) + \epsilon$, where $\theta \sim p_\theta^{(\tilde{c})}(\theta)$ is a discrete distribution on a finite set $\Theta^{(\tilde{c})} \subseteq \Theta$, and an ANM in the anti-causal direction $X = g(Y;\omega = \omega_{\tilde{c}}) + \tilde{\epsilon}$. According to lemma 1, for each data block, to ensure the existence of an ANM-MM in the causal direction and an ANM in the anti-causal direction, $(p_X, p_\epsilon, f, p_\theta)$ should fulfill an ordinary differential equation in the form of (2). Then the existence of backward ANM-MM requires $\tilde{C}$ ordinary differential equations to be fulfilled simultaneously which yields (3). $\qquad\square$

Then the causal direction in ANM-MM can be inferred by investigating the independence between the hypothetical cause and the corresponding function parameter. According to theorem 1, if they are independent in the causal direction, then it is highly likely they are dependent in the anticausal direction. Therefore in practice, the inferred direction is the one that shows more evidence of independence between them.

## 2.3 Mechanism clustering of ANM-MM

In ANM-MM, $\theta$, which represents function parameters, can be directly used to identify different g.m.s since each parameter value corresponds to one mechanism. In other words, observations generated by the same g.m. would have the same $\theta$ if the imposed statistical model is identifiable with respect to $\theta$.

Denote the parameter associated with each observation $(x_n, y_n)$ by $\boldsymbol{\theta}_n$, we suppose a more practical inherent clustering structure behind hidden $\boldsymbol{\theta}_n$. Formally, there is a grouping indicator of integers $\mathbf{z} \in \{1, \ldots, C\}^N$ that assign each $\boldsymbol{\theta}_n$ to one of the $C$ clusters, through the $n$th element of $\mathbf{z}$, e.g. $\boldsymbol{\theta}_n$ belongs to cluster $c$ if $[\mathbf{z}]_n = c, c \in \{1, \ldots, C\}$. Following ANM-MM, we may assume each $\boldsymbol{\theta}_n$ belong to one of $C$ components and each component follows $\mathcal{N}(\mu_c, \sigma^2)$. A likelihood-based clustering scheme suggests minimizing $-\ell$ jointly with respect to all means and $\mathbf{z}$

$$\ell(\mathcal{M}, \mathbf{z}) = \log \prod_{n=1}^N \prod_{c=1}^C \left\{ \frac{1}{\sqrt{2\pi}\sigma} \exp\left( -\frac{1}{2\sigma^2} (\boldsymbol{\theta}_n - \mu_c)^2 \right) \right\}^{\mathbf{1}([\mathbf{z}]_n = c)},$$

where $\mathcal{M} = \{\mu_c\}_{c=1}^C$ and $\mathbf{1}(\cdot)$ is the indicator function. To simplify further let's ignore the known $\sigma^2$ and minimize $-\ell$ using coordinate descent iteratively

$$\hat{\mathcal{M}} \mid \mathbf{z} = \underset{\mathcal{M}}{\arg\min} \sum_{c=1}^C \sum_{\{n|[\mathbf{z}]_n=c\}} (\boldsymbol{\theta}_n - \mu_c)^2 \tag{4}$$

$$\hat{\mathbf{z}} \mid \mathcal{M} = \underset{\mathbf{z}}{\arg\min} \sum_{c=1}^C \sum_{\{n|[\mathbf{z}]_n=c\}} (\boldsymbol{\theta}_n - \mu_c)^2. \tag{5}$$

The minimizer of (4) is the mean $\hat{\mu}_c = \frac{1}{n_c} \sum_{\{n|[\mathbf{z}]_n=c\}} \boldsymbol{\theta}_n$, where $n_c$ is the size of the $c$th cluster $n_c = \sum_{n=1}^N \mathbf{1}([\mathbf{z}]_n = c)$. The minimizer of (5) is group assignment through minimum Euclidean distance. Therefore, iterating between (4) and (5) coincides with applying $k$-means algorithm on all $\boldsymbol{\theta}_n$ and the goal of finding clusters consistent with the g.m.s for data from ANM-MM can be achieved by firstly estimating parameters associated with each observation and then conducting $k$-means directly on parameters.

# 3 ANM-MM Estimation by GPPOM

We propose Gaussian process partially observable model (GPPOM) and incorporate Hilbert-Schmidt independence criterion (HSIC) [4] enforcement into GPPOM to estimate the model parameter $\theta$. Then we summarize algorithms for causal inference and mechanism clustering of ANM-MM.

## 3.1 Preliminaries

**Dual PPCA.** Dual PPCA [11] is a latent variable model in which maximum likelihood solution for the latent variables is found by marginalizing out the parameters. Given a set of $N$ centered $D$-dimensional data $\mathbf{Y} = [\boldsymbol{y}_1, \ldots, \boldsymbol{y}_N]^T$, dual PPCA learns the $q$-dimensional latent representation $\boldsymbol{x}_n$ associated with each observation $\boldsymbol{y}_n$. The relation between $\boldsymbol{x}_n$ and $\boldsymbol{y}_n$ in dual PPCA is $\boldsymbol{y}_n = \mathbf{W}\boldsymbol{x}_n + \boldsymbol{\epsilon}_n$, where the matrix $\mathbf{W}$ specifies the linear relation between $\boldsymbol{y}_n$ and $\boldsymbol{x}_n$ and noise $\boldsymbol{\epsilon}_n \sim \mathcal{N}(\mathbf{0}, \beta^{-1}\mathbf{I})$. Then by placing a standard Gaussian prior on each row of $\mathbf{W}$, one obtains the marginal likelihood of all observations and the objective function of dual PPCA is the log-likelihood $\mathcal{L} = -\frac{DN}{2}\ln(2\pi) - \frac{D}{2}\ln(|\mathbf{K}|) - \frac{1}{2}\operatorname{tr}\left(\mathbf{K}^{-1}\mathbf{Y}\mathbf{Y}^T\right)$, where $\mathbf{K} = \mathbf{X}\mathbf{X}^T + \beta^{-1}\mathbf{I}$ and $\mathbf{X} = [\boldsymbol{x}_1, \ldots, \boldsymbol{x}_N]^T$.

**GP-LVM.** GP-LVM [10] generalizes dual PPCA to cases of nonlinear relation between $\boldsymbol{y}_n$ and $\boldsymbol{x}_n$ by mapping latent representations in $\mathbf{X}$ to a feature space, i.e. $\boldsymbol{\Phi} = [\phi(\boldsymbol{x}_1), \ldots, \phi(\boldsymbol{x}_N)]^T$, where $\phi(\cdot)$ denotes the canonical feature map. Then $\mathbf{K} = \boldsymbol{\Phi}\boldsymbol{\Phi}^T + \beta^{-1}\mathbf{I}$ and $\boldsymbol{\Phi}\boldsymbol{\Phi}^T$ can be computed using kernel trick. GP-LVM can also be interpreted as a new class of models which consists of $D$ independent Gaussian processes [19] mapping from a latent space to an observed data space [10].

**HSIC.** HSIC [4], which is based on reproducing kernel Hilbert space (RKHS) theory, is widely used to measure the dependence between r.v.s. Let $\mathcal{D} := \{(\boldsymbol{x}_n, \boldsymbol{y}_n)\}_{n=1}^N$ be a sample of size $N$ draw independently and identically distributed according to $P(X, Y)$, HSIC answers the query whether $X \perp\!\!\!\perp Y$. Formally, denote by $\mathcal{F}$ and $\mathcal{G}$ RKHSs with universal kernel $k, l$ on the compact domains $\mathcal{X}$ and $\mathcal{Y}$, HSIC is the measure defined as $\operatorname{HSIC}(P(X,Y), \mathcal{F}, \mathcal{G}) := \|\mathcal{C}_{xy}\|_{\mathrm{HS}}^2$, which is essentially the squared Hilbert Schmidt norm [4] of the cross-covariance operator $\mathcal{C}_{xy}$ from RKHS $\mathcal{G}$ to $\mathcal{F}$ [3]. It is proved in [4] that, under conditions specified in [5], $\operatorname{HSIC}(P(X,Y), \mathcal{F}, \mathcal{G}) = 0$ if and only if $X \perp\!\!\!\perp Y$. In practice, a biased empirical estimator of HSIC based on the sample $\mathcal{D}$ is often adopted:

$$\operatorname{HSIC}_b(\mathcal{D}) = \frac{1}{N^2}\operatorname{tr}(\mathbf{K}\mathbf{H}\mathbf{L}\mathbf{H}), \tag{6}$$

where $[\mathbf{K}]_{ij} = k(\boldsymbol{x}_i, \boldsymbol{x}_j)$, $[\mathbf{L}]_{ij} = l(\boldsymbol{y}_i, \boldsymbol{y}_j)$, $\mathbf{H} = \mathbf{I} - \frac{1}{N}\vec{\mathbf{1}}\vec{\mathbf{1}}^T$, and $\vec{\mathbf{1}}$ is a $N \times 1$ vector of ones.

## 3.2 Gaussian process partially observable model

**Partially observable dual PPCA.** Dual PPCA is not directly applicable to model ANM-MM since: 1) part of the r.v. that maps to the effect is visible (i.e. $X$); 2) the relation (i.e. $f$) is nonlinear; 3) r.v.s that contribute to the effect should be independent ($X \perp\!\!\!\perp \theta$) in ANM-MM. To tackle 1), a latent r.v. $\theta$ is brought in dual PPCA.

Denote the observed effect by $\mathbf{Y} = [\boldsymbol{y}_1, \ldots, \boldsymbol{y}_N]^T$, observed cause by $\mathbf{X} = [\boldsymbol{x}_1, \ldots, \boldsymbol{x}_N]^T$, the matrix collecting function parameters associated with each observation by $\boldsymbol{\Theta} = [\boldsymbol{\theta}_1, \ldots, \boldsymbol{\theta}_N]^T$ and the r.v. that contribute to the effect by $\tilde{X} = [X, \theta]$. Similar to dual PPCA, the relation between the latent representation and the observation is given by

$$\boldsymbol{y}_n = \tilde{\mathbf{W}}\tilde{\boldsymbol{x}}_n + \boldsymbol{\epsilon}_n, \quad n = 1, \ldots, N$$

where $\tilde{\boldsymbol{x}}_n = \left[\boldsymbol{x}_n^T, \boldsymbol{\theta}_n^T\right]^T$, $\tilde{\mathbf{W}}$ is the matrix specifies the relation between $\boldsymbol{y}_n$ and $\tilde{\boldsymbol{x}}_n$, $\boldsymbol{\epsilon}_n \sim \mathcal{N}(\mathbf{0}, \beta^{-1}\mathbf{I})$ is the additive noise. Then by placing a standard Gaussian prior on $\tilde{\mathbf{W}}$, i.e. $p(\tilde{\mathbf{W}}) = \prod_{i=1}^D \mathcal{N}(\tilde{\mathbf{w}}_{i,:}|\mathbf{0}, \mathbf{I})$, where $\tilde{\mathbf{w}}_{i,:}$ is the $i$th row of the matrix $\tilde{\mathbf{W}}$, the log-likelihood of the observations is given by

$$\mathcal{L}(\boldsymbol{\Theta}|\mathbf{X}, \mathbf{Y}, \beta) = -\frac{DN}{2}\ln(2\pi) - \frac{D}{2}\ln\left(|\tilde{\mathbf{K}}|\right) - \frac{1}{2}\operatorname{tr}\left(\tilde{\mathbf{K}}^{-1}\mathbf{Y}\mathbf{Y}^T\right), \tag{7}$$

where $\tilde{\mathbf{K}} = \tilde{\mathbf{X}}\tilde{\mathbf{X}}^T + \beta^{-1}\mathbf{I} = [\mathbf{X}, \boldsymbol{\Theta}][\mathbf{X}, \boldsymbol{\Theta}]^T + \beta^{-1}\mathbf{I} = \mathbf{X}\mathbf{X}^T + \boldsymbol{\Theta}\boldsymbol{\Theta}^T + \beta^{-1}\mathbf{I}$ is the covariance matrix after bringing in $\theta$.

**General nonlinear cases (GPPOM).** Similar to the generalization from dual PPCA to GP-LVM, the dual PPCA with observable $X$ and latent $\theta$ can be easily generalized to nonlinear cases. Denote the feature map by $\phi(\cdot)$ and $\boldsymbol{\Phi} = [\phi(\tilde{\boldsymbol{x}}_1), \ldots, \phi(\tilde{\boldsymbol{x}}_N)]^T$, then the covariance matrix is given by $\tilde{\mathbf{K}} = \boldsymbol{\Phi}\boldsymbol{\Phi}^T + \beta^{-1}\mathbf{I}$. The entries of $\boldsymbol{\Phi}\boldsymbol{\Phi}^T$ can be computed using kernel trick given a selected kernel $k(\cdot, \cdot)$. In this paper, we adopt the radial basis function (RBF) kernel, which reads $k(\boldsymbol{x}_i, \boldsymbol{x}_j) = \exp\left(-\sum_{d=1}^{D_x} \gamma_d(\boldsymbol{x}_{id} - \boldsymbol{x}_{jd})^2\right)$, where $\gamma_d$, for $d = 1, \ldots, D_x$, are free parameters and $D_x$ is the dimension of the input. As a result of adopting RBF kernel, the covariance matrix $\tilde{\mathbf{K}}$ in (7) can be computed as

$$\tilde{\mathbf{K}} = \boldsymbol{\Phi}\boldsymbol{\Phi}^T + \beta^{-1}\mathbf{I} = \mathbf{K}_X \circ \mathbf{K}_\theta + \beta^{-1}\mathbf{I},$$

---

**Algorithm 1:** Causal Inference

**input** : $\mathcal{D} = \{(\boldsymbol{x}_n, \boldsymbol{y}_n)\}_{n=1}^N$ - the set of observations of two r.v.s; $\lambda$ - parameter of independence

**output :** The causal direction

1 Standardize observations of each r.v.;
2 Initialize $\beta$ and kernel parameters;
3 Optimize (8) in both directions, denote the value of HSIC term by $\text{HSIC}_{X \to Y}$ and $\text{HSIC}_{Y \to X}$, respectively;
4 **if** *$\text{HSIC}_{X \to Y} < \text{HSIC}_{Y \to X}$* **then**
5    The causal direction is $X \to Y$;
6 **else if** *$\text{HSIC}_{X \to Y} > \text{HSIC}_{Y \to X}$* **then**
7    The causal direction is $Y \to X$;
8 **else**
9    No decision made.
10 **end**

---

where $\circ$ denotes the Hadamard product, the entries on $i$th row and $j$th column of $\mathbf{K}_X$ and $\mathbf{K}_\theta$ are given by $[\mathbf{K}_X]_{ij} = k(\boldsymbol{x}_i, \boldsymbol{x}_j)$ and $[\mathbf{K}_\theta]_{ij} = k(\boldsymbol{\theta}_i, \boldsymbol{\theta}_j)$, respectively. After the nonlinear generalization, the relation between $Y$ and $\tilde{X}$ reads $Y = f(\tilde{X}) + \epsilon = f(X, \theta) + \epsilon$. This variant of GP-LVM with partially observable latent space is named GPPOM in this paper. Like GP-LVM, $\tilde{X}$ is mapped to $Y$ by the same set of Gaussian processes in GPPOM so the differences in the g.m.s is captured by $\boldsymbol{\theta}_n$, the latent representations associated with each observation.

## 3.3 Model estimation by independence enforcement

Both dual PPCA and GP-LVM finds the latent representations through log-likelihood maximization using scaled conjugate gradient [14]. However, the $\boldsymbol{\theta}$ can not be found by directly conducting likelihood maximization since the ANM-MM requires additionally the independence between $X$ and $\boldsymbol{\theta}$. To this end, we include HSIC [4] in the objective. By incorporating HSIC term into the negative log-likelihood of GPPOM, the optimization objective reads

$$\arg\min_{\boldsymbol{\Theta}, \Omega} \mathcal{J}(\boldsymbol{\Theta}) = \arg\min_{\boldsymbol{\Theta}, \Omega} \left[ -\mathcal{L}(\boldsymbol{\Theta}|\mathbf{X}, \mathbf{Y}, \Omega) + \lambda \log \text{HSIC}_b(\mathbf{X}, \boldsymbol{\Theta}) \right], \quad (8)$$

where $\lambda$ is the parameter which controls the importance of the HSIC term and $\Omega$ is the set of all hyper parameters including $\beta$ and all kernel parameters $\gamma_d$, $d = 1, \ldots, D_x$.

To find $\boldsymbol{\Theta}$, we resort to the gradient descant methods. The gradient of the objective $\mathcal{J}$ with respect to latent points in $\boldsymbol{\Theta}$ is given by

$$\frac{\partial \mathcal{J}}{\partial [\boldsymbol{\Theta}]_{ij}} = \text{tr}\left[ \left(\frac{\partial \mathcal{J}}{\partial \mathbf{K}_\theta}\right)^T \frac{\partial \mathbf{K}_\theta}{\partial [\boldsymbol{\Theta}]_{ij}} \right]. \quad (9)$$

The first part on the right hand side of (9), which is the gradient of $\mathcal{J}$ with respect to the kernel matrix $\mathbf{K}_\theta$, can be computed as

$$\frac{\partial \mathcal{J}}{\partial \mathbf{K}_\theta} = -\text{tr}\left[ \left(\tilde{\mathbf{K}}^{-1}\mathbf{Y}\mathbf{Y}^T\tilde{\mathbf{K}}^{-1} - D\tilde{\mathbf{K}}^{-1}\right)^T \left(\mathbf{K}_X \circ \mathbf{J}^{ij}\right) \right] + \lambda \frac{1}{\text{tr}\left(\mathbf{K}_X\mathbf{H}\mathbf{K}_\theta\mathbf{H}\right)} \mathbf{H}\mathbf{K}_X\mathbf{H}, \quad (10)$$

where $\mathbf{J}^{ij}$ is the single-entry matrix, 1 at $(i, j)$ and 0 elsewhere and $\mathbf{H} = \mathbf{I} - \frac{1}{N}\vec{\mathbf{1}}\vec{\mathbf{1}}^T$. Combining $\frac{\partial \mathcal{L}}{\partial \mathbf{K}_\theta}$ with $\frac{\partial \mathbf{K}_\theta}{\partial [\boldsymbol{\Theta}]_{ij}}$, whose entry on the $m$th row and $n$th column is given by $\frac{\partial [\mathbf{K}_\Theta]_{mn}}{\partial [\boldsymbol{\Theta}]_{ij}} = \frac{\partial k(\boldsymbol{\theta}_m, \boldsymbol{\theta}_n)}{\partial [\boldsymbol{\Theta}]_{ij}}$, through the chain rule, all latent points in $\boldsymbol{\Theta}$ can be optimized. With $\boldsymbol{\Theta}$, one can conduct causal inference and mechanism clustering of ANM-MM. The detailed steps are given in Algorithm 1 and 2.

## 4 Experiments

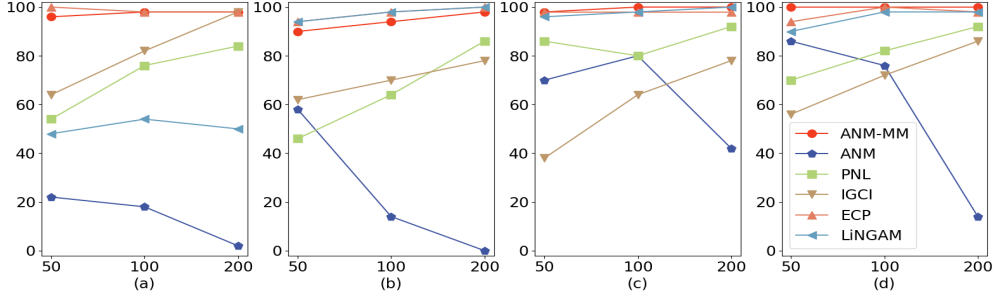

Figure 3: Accuracy ($y$-axis) versus sample size ($x$-axis) on $Y = f(X; \theta_c) + \epsilon$ with different mechanisms. (a) $f_1$, (b) $f_2$, (c) $f_3$, (d) $f_4$.

In this section, experimental results on both synthetic and real data are given to show the performance of ANM-MM on causal inference and mechanism clustering tasks. The Python code of ANM-MM is available online at `https://github.com/amber0309/ANM-MM`.

### 4.1 Synthetic data

In experiments of causal inference, ANM-MM is compared with ANM [6], PNL [21], IGCI [8], ECP [20] and LiNGAM [18]. The results are evaluated using accuracy, which is the percentage of correct causal direction estimation

---

**Algorithm 2:** Mechanism clustering

**input** : $\mathcal{D} = \{(\boldsymbol{x}_n, \boldsymbol{y}_n)\}_{n=1}^{N}$ - the set of observations of two r.v.s;
$\lambda$ - parameter of independence;
$C$ - Number of clusters

**output :** The cluster labels

1 Standardize observations of each r.v.;
2 Initialize $\beta$ and kernel parameters;
3 Find $\Theta$ by optimizing (8) in causal direction;
4 Apply $k$-means on $\boldsymbol{\theta}_n, n = 1, \ldots, N$;
5 **return** the cluster labels.

---

of 50 independent experiments. Note that ANM-MM was applied using different parameter $\lambda \in \{0.001, 0.01, 0.1, 1, 10\}$ and IGCI was applied using different reference measures and estimators. Their highest accuracy is reported.

In experiments of clustering, ANM-MM is compared with well-known $k$-means [13] (similarity-based) on both raw data ($k$-means) and its PCA component (PCA-$k$m), Gaussian mixture clustering (GMM) [16] (model-based), spectral clustering (SpeClu) [17] (spectral graph theory-based) and DBSCAN [2] (density-based). Clustering performance is evaluated using average adjusted Rand index [7] (avgARI), which is the mean ARI over 100 experiments. High ARI ($\in [-1, 1]$) indicates good match between the clustering results and the ground truth. Sample size ($N$) is 100 in all synthetic clustering experiments. Clustering results are visualized in the supplementary[1].

**Different g.m.s and sample sizes.** We examine the performance on different g.m.s ($f$) and sample sizes ($N$). The mechanisms adopted are the following elementary functions: 1) $f_1 = \frac{1}{1.5 + \theta_c X^2}$; 2) $f_2 = 2 \times X^{\theta_c - 0.25}$; 3) $f_3 = \exp(-\theta_c X)$; 4) $f_4 = \tanh(\theta_c X)$. We tested sample size $N = 50, 100$ and 200 for each mechanism. Given $f$ and $N$, the cause $X$ is sampled from a uniform distribution $U(0, 1)$ and then mapped to the effect by $Y = f(X; \theta_c) + \epsilon, c \in \{1, 2\}$, where $\theta_1 \sim U(1, 1.1)$, $\theta_2 \sim U(3, 3.1)$ and $\epsilon \sim \mathcal{N}(0, 0.05^2)$. Each mechanism generates half of the observations.

*Causal Inference.* The results are shown in Fig. 3. ANM-MM and ECP outperforms others based on a single causal model, which is consistent with our anticipation. Compared with ECP, ANM-MM shows slight advantages in 3 out of 4 settings. *Clustering.* The avgARI values are summarized in (i) of Table 1. ANM-MM significantly outperforms other approaches in all mechanism settings.

**Different number of g.m.s.**[2] We examine the performance on different number of g.m.s ($C$ in Definition 1). $\theta_1$, $\theta_2$ and $\epsilon$ are the same as in previous experiments. In the setting of three mechanisms, $\theta_3 \sim U(0.5, 0.6)$. In the setting of four, $\theta_3 \sim U(0.5, 0.6)$ and $\theta_4 \sim U(2, 2.1)$. Again, the numbers of observations from each mechanism are the same.

Table 1: avgARI of synthetic clustering experiments

| avgARI | (i) $f$ | | | | (ii) $C$ | | (iii) $\sigma$ | | (iv) $a_1$ | |
|---|---|---|---|---|---|---|---|---|---|---|
| | $f_1$ | $f_2$ | $f_3$ | $f_4$ | 3 | 4 | 0.01 | 0.1 | 0.25 | 0.75 |
| ANM-MM | **0.393** | **0.660** | **0.777** | **0.682** | **0.610** | **0.447** | **0.798** | **0.608** | **0.604** | **0.867** |
| $k$-means | 0.014 | 0.039 | 0.046 | 0.046 | 0.194 | 0.165 | 0.049 | 0.042 | 0.047 | 0.013 |
| PCA-$k$m | 0.013 | 0.037 | 0.044 | 0.048 | 0.056 | 0.041 | 0.047 | 0.040 | 0.052 | 0.014 |
| GMM | 0.015 | 0.340 | 0.073 | 0.208 | 0.237 | 0.202 | 0.191 | 0.025 | 0.048 | 0.381 |
| SpeClu | 0.003 | 0.129 | 0.295 | 0.192 | 0.285 | 0.175 | 0.595 | 0.048 | 0.044 | -0.008 |
| DBSCAN | 0.055 | 0.265 | 0.342 | 0.358 | 0.257 | 0.106 | 0.527 | 0.110 | 0.521 | 0.718 |

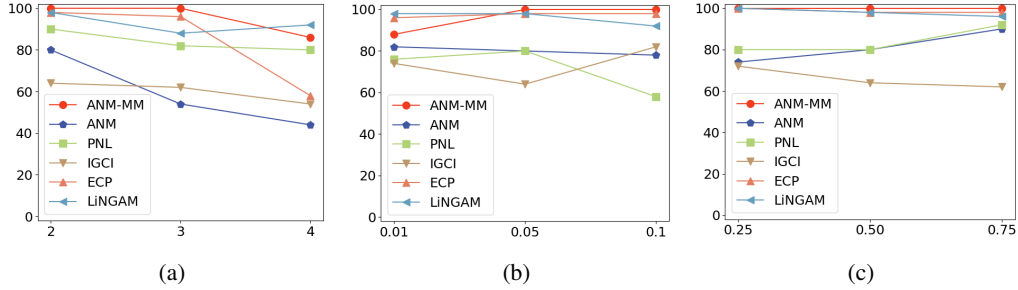

      (a)                          (b)                          (c)

Figure 4: Accuracy (y-axis) versus (a) number of mechanisms; (b) noise standard deviation; (c) mixing proportion; on $f_3$ with $N = 100$.

*Causal Inference.* The results are given in Fig. 4a which shows decreasing trend for all approaches. However, ANM-MM keeps 100% when the number of mechanisms increases from 2 to 3. *Clustering.* The avgARI values are given in (ii) and (i)$f_3$ of Table 1. The performance of different approaches show different trends which is probably due to the clustering principle they are based on. Although ANM-MM is heavily influenced by $C$, its performance is still much better than others.

**Different noise standard deviations.** We examine the performance on different noise standard deviations $\sigma$. $\theta_1, \theta_2$ are the same as in the first part of experiments. Three different cases where $\sigma = 0.01, 0.05$ and $0.1$ are tested.

*Causal Inference.* The results are given in Fig. 4b. The change in $\sigma$ in this range does not significantly influence the performance of most causal inference approaches. ANM-MM keeps 100% accuracy for all choice of $\sigma$. *Clustering.* The avgARI values are given in (iii) and (i)$f_3$ of Table 1. As our anticipation, the clustering results heavily rely on $\sigma$ and all approaches show a decreasing trend in avgARI as $\sigma$ increases. However, ANM-MM is the most robust against large $\sigma$.

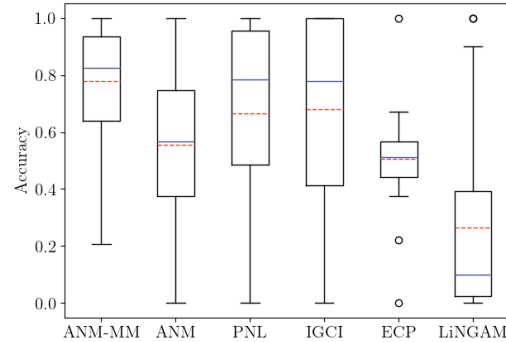

Figure 5: Accuracy on real cause-effect pairs.

**Different mixing proportions.** We examine the performance on different mixing proportions ($a_c$ in Definition 1). $\theta_1, \theta_2$ and $\sigma$ are the same as in the first part of experiments. Cases where $a_1 = 0.25, 0.5$ and $0.75$ (corresponding $a_2 = 0.75, 0.5$ and $0.25$) are tested.

*Causal Inference.* The results on different $a_1$ are given in Fig. 4c. Approaches based on a single causal model are sensitive to the change in $a_1$ whereas ECP and ANM-MM are more robust and outperform others. *Clustering.* The avgARI values of experiments on different $a_1$ are given in (iv) and (i)$f_3$ of Table 1. The results of comparing approaches are significantly affected by $a_1$ and ANM-MM shows best robustness against the change in $a_1$.

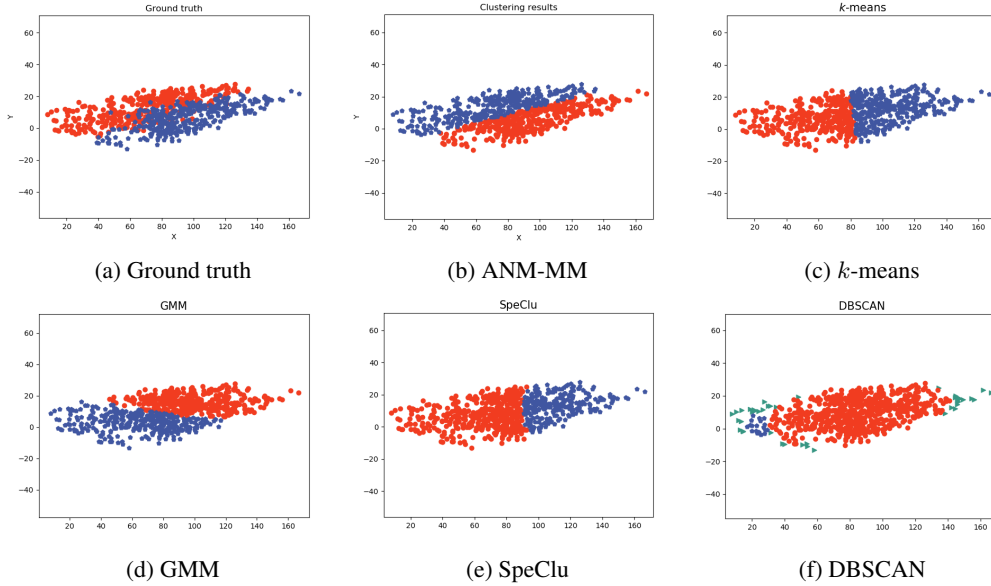

| (a) Ground truth | (b) ANM-MM | (c) $k$-means |
| (d) GMM | (e) SpeClu | (f) DBSCAN |

Figure 6: Ground truth and clustering results of different approaches on BAFU air data.

## 4.2 Real data

**Causal inference on Tüebingen cause-effect pairs.** We evaluate the causal inference performance of ANM-MM on real world benchmark cause-effect pairs[3] [15]. Nine out of 41 data sets are excluded in our experiment because either they consists of multivariate or categorical data (pair 47, 52, 53, 54, 55, 70, 71, 101 and 105) or the estimated latent representations are extremely close[4] (pair 12 and 17). Fifty independent experiments are repeated for each pair, and the percentage of correct inference of different approaches are recorded. Then average percentage of pairs from the same data set is computed as the accuracy of the corresponding data set. In each independent experiment, different inference approaches are applied on 90 points randomly sampled from raw data without replacement.

The results are summarized in Fig. 5 with blue solid line indicating median accuracy and red dashed line indicating mean accuracy. It shows that the performance of ANM-MM is satisfactory, with highest median accuracy of about 82%. IGCI also performs quite well, especially in terms of median, followed by PNL.

**Clustering on BAFU air data.** We evaluate the clustering performance of ANM-MM on real air data obtained online[5]. This data consists of daily mean values of ozone ($\mu g/m^3$) and temperature (°) of 2009 from two distinct locations in Switzerland. In our experiment, we regard the data as generating from two mechanisms (each corresponds to a location). The clustering results are visualized in Fig. 6. The ARI values of ANM-MM is 0.503, whereas $k$-means, GMM, spectral clustering and DBSCAN could only obtain ARI of -0.001, 0.003, 0.078 and 0.003, respectively. ANM-MM is the only one that could reveal the property related to the location of the data g.m..

## 5 Conclusion

In this paper, we extend the ANM to a more general model (ANM-MM) in which there are a finite number of ANMs of the same function form and differ only in parameter values. The condition of identifiability of ANM-MM is analyzed. To estimate ANM-MM, we adopt the GP-LVM framework and propose a variant of it called GPPOM to find the optimized latent representations and further conduct causal inference and mechanism clustering. Results on both synthetic and real world data verify the effectiveness of our proposed approach.

**Acknowledgments**

This work is partially supported by the Hong Kong Research Grants Council.

## Footnotes

[1]The results of PCA-$k$m are not visualized since they are similar to and worse than those of $k$-means.

[2]From this part on, g.m. is fixed to be $f_3$.

[3] https://webdav.tuebingen.mpg.de/cause-effect/.

[4] close in the sense that $|\theta_i - \theta_j| < 0.001$.

[5] https://www.bafu.admin.ch/bafu/en/home/topics/air.html

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
