[Supplementary Material · supplementary_for_ANM_MM.pdf]

# Supplementary for ANM-MM

**Shoubo Hu**[*], **Zhitang Chen**[†], **Vahid Partovi Nia**[†], **Laiwan Chan**[*], **Yanhui Geng**[‡]

[*]The Chinese University of Hong Kong; [†]Huawei Noah's Ark Lab;
[‡]Huawei Montréal Research Center
[*]{sbhu, lwchan}@cse.cuhk.edu.hk
[†‡]{chenzhitang2, vahid.partovinia, geng.yanhui}@huawei.com

## Overview

- Proof of Lemma 1
- Derivation of (10)
- Adjusted Rand Index
- Clustering Results Visualization

## A  Proof of Lemma 1

*Proof.* If there exists an Additive Noise Model (ANM) in the backward direction, i.e.,

$$X = g(Y) + \tilde{\epsilon},$$

where $\tilde{\epsilon} \perp\!\!\!\perp Y$, then we have

$$p(X, Y) = p_{\tilde{\epsilon}}(X - g(Y))p_Y(Y),$$

and thus

$$\pi(X, Y) = \log p(X, Y) = \log(p_{\tilde{\epsilon}}(X - g(Y))) + \log p_Y(Y).$$

Denote by $\tilde{v}(\cdot) = \log p_{\tilde{\epsilon}}(\cdot)$ and $\tilde{\xi}(\cdot) = \log p_Y(\cdot)$. Taking partial derivative of $\pi(X, Y)$ with respect to $X$, we get

$$\frac{\partial \pi}{\partial X} = \tilde{v}'(X - g(Y)).$$

Furthermore, we have

$$\frac{\partial^2 \pi}{\partial X^2} = \tilde{v}''(X - g(Y)),$$

and

$$\frac{\partial \pi}{\partial X \partial Y} = -\tilde{v}''(X - g(Y))g'(Y).$$

We find that

$$\frac{\partial^2 \pi / \partial X \partial Y}{\partial \pi / \partial^2 X} = -g'(Y),$$

and thus

$$\frac{\partial}{\partial X} \left( \frac{\partial^2 \pi / \partial X \partial Y}{\partial^2 \pi / \partial X^2} \right) = 0.$$

Let us get back to the forward model where we have

$$p(X, Y) = p_X(X) \sum_{c=1}^{C} a_c p_{\epsilon}(Y - f_c(X)). \tag{1}$$

Taking $\log$ of both sides of (1), we get

$$\pi(X,Y) = \log p(X,Y) = \log \sum_{c=1}^{C} a_c p_\epsilon(Y - f_c(X)) + \log p_X(X).$$

For notation simplicity, we drop the argument of $p_\epsilon(Y - f_c(X))$ and denote by $\xi(\cdot) = \log(p_X(\cdot))$, we get

$$\frac{\partial \pi}{\partial X} = \frac{-1}{\sum_c a_c p_\epsilon(Y - f_c(X))} \sum_c a_c p'_\epsilon(Y - f_c(X)) f'_c(X) + \xi'(X)$$

and

$$\frac{\partial^2 \pi}{\partial X \partial Y} = \frac{1}{\left(\sum_c a_c p_\epsilon(Y - f_c(X))\right)^2} \sum_c a_c p'_\epsilon(Y - f_c(X)) \sum_c a_c p'_\epsilon(Y - f_c(X)) f'_c(X)$$
$$+ \frac{-1}{\sum_c a_c p_\epsilon(Y - f_c(X))} \sum_c a_c p''_\epsilon(Y - f_c(X)) f'_c(X)$$

$$\frac{\partial^2 \pi}{\partial X^2} = \frac{-1}{\left(\sum_c a_c p_\epsilon(Y - f_c(X))\right)^2} \left(\sum_c a_c p'_\epsilon(Y - f_c(X)) f'_c(X)\right)^2$$
$$+ \frac{1}{\sum_c a_c p_\epsilon(Y - f_c(X))} \sum_c a_c p''_\epsilon(Y - f_c(X))(f'_c(X))^2$$
$$+ \frac{-1}{\sum_c a_c p_\epsilon(Y - f_c(X))} \sum_c a_c p'_\epsilon(Y - f_c(X)) f''_c(X) + \xi''(X)$$

Let

$$u = \frac{\partial^2 \pi}{\partial X \partial Y}$$

and denote by $p_{\epsilon,c} = p_\epsilon(Y - f_c(X))$, $p'_{\epsilon,c} = p'_\epsilon(Y - f_c(X))$, $p''_{\epsilon,c} = p''_\epsilon(Y - f_c(X))$, $p'''_{\epsilon,c} = p'''_\epsilon(Y - f_c(X))$ and $f_c = f_c(X)$, $f'_c = f'_c(X)$, $f''_c = f''_c(X)$ and $f'''_c = f'''_c(X)$, $\xi = \xi(X)$, $\xi' = \xi'(X)$, $\xi'' = \xi''(X)$ and $\xi''' = \xi'''(X)$. We have

$$\frac{\partial u}{\partial X} = \frac{2}{\left(\sum_c a_c p_{\epsilon,c}\right)^3} \sum_c a_c p_{\epsilon,c} f'_c \sum_c a_c p'_{\epsilon,c} \sum_c a_c p'_{\epsilon,c} f'_c$$
$$+ \frac{1}{\left(\sum_c a_c p_{\epsilon,c}\right)^2} \left(-\sum_c a_c p''_{\epsilon,c} f'_c \sum_c a_c p'_{\epsilon,c} f'_c - \sum_c a_c p'_{\epsilon,c} \sum_c a_c p''_{\epsilon,c}(f'_c)^2 + \sum_c a_c p'_{\epsilon,c} \sum_c a_c p'_{\epsilon,c} f''_c\right)$$
$$+ \frac{-1}{\left(\sum_c a_c p_{\epsilon,c}\right)^2} \sum_c a_c p'_{\epsilon,c} f'_c \sum_c a_c p''_{\epsilon,c} f'_c + \frac{1}{\sum_c a_c p_{\epsilon,c}} \sum_c a_c p'''_{\epsilon,c}(f'_c)^2 + \frac{1}{\sum_c a_c p_{\epsilon,c}} \sum_c a_c p''_{\epsilon,c} f''_c$$
$$= \frac{2}{\left(\sum_c a_c p_{\epsilon,c}\right)^3} \sum_c a_c p_{\epsilon,c} f'_c \sum_c a_c p'_{\epsilon,c} \sum_c a_c p'_{\epsilon,c} f'_c$$
$$+ \frac{1}{\left(\sum_c a_c p_{\epsilon,c}\right)^2} \left(-2\sum_c a_c p''_{\epsilon,c} f'_c \sum_c a_c p'_{\epsilon,c} f'_c - \sum_c a_c p'_{\epsilon,c} \sum_c a_c p''_{\epsilon,c}(f'_c)^2 + \sum_c a_c p'_{\epsilon,c} \sum_c a_c p'_{\epsilon,c} f''_c\right)$$
$$+ \frac{1}{\sum_c a_c p_{\epsilon,c}} \sum_c a_c p'''_{\epsilon,c}(f'_c)^2 + \frac{1}{\sum_c a_c p_{\epsilon,c}} \sum_c a_c p''_{\epsilon,c} f''_c.$$

Denote by

$$v = \frac{\partial^2 \pi}{\partial X^2},$$

then we have

$$
\frac{\partial v}{\partial X} = \frac{-2}{(\sum_c a_c p_{\epsilon,c})^3}\left(\sum_c a_c p'_{\epsilon,c} f'_c\right)^3 + \frac{-2}{(\sum_c a_c p_{\epsilon,c})^2}(\sum_c a_c p'_{\epsilon,c} f'_c)\sum_c a_c(p''_{\epsilon,c}(-f'_c)f_c + p'_{\epsilon,c}f''_c)
$$

$$
+ \frac{-1}{(\sum_c a_c p_{\epsilon,c})^2}\sum_c a_c p'_{\epsilon,c}f'_c \sum_c a_c p''_{\epsilon,c}(f'_c)^2 + \frac{-1}{\sum_c a_c p_{\epsilon,c}}\sum_c a_c p'''_{\epsilon,c}(f_c)^3 + \frac{2}{\sum_c a_c p_{\epsilon,c}}\sum_c a_c p''_{\epsilon,c} f'_c f''_c
$$

$$
+ \frac{-1}{(\sum_c a_c p_{\epsilon,c})^2}\sum_c a_c p'_{\epsilon,c}f'_c \sum_c a_c p'_{\epsilon,c}f''_c + \frac{1}{\sum_c a_c p_{\epsilon,c}}\sum_c a_c p''_{\epsilon,c} f'_c f''_c + \frac{-1}{\sum_c a_c p_{\epsilon,c}}\sum_c a_c p'_{\epsilon,c} f'''_c + \xi'''
$$

Further denote by

$$
U(X,Y) = \frac{\partial^2 \pi}{\partial X \partial Y} = \frac{1}{(\sum_c a_c p_{\epsilon,c})^2}\sum_c a_c p'_{\epsilon,c}\sum_c a_c p'_{\epsilon,c}f'_c + \frac{-1}{\sum_c a_c p_{\epsilon,c}}\sum_c a_c p''_{\epsilon,c}f'_c
$$

and

$$
V(X,Y) = \frac{\partial^2 \pi}{\partial X^2} = \frac{-1}{(\sum_c a_c p_{\epsilon,c})^2}\left(\sum_c a_c p'_{\epsilon,c}f'_c\right)^2 + \frac{1}{\sum_c a_c p_{\epsilon,c}}\sum_c a_c p''_{\epsilon,c}(f'_c)^2 + \frac{-1}{\sum_c a_c p_{\epsilon,c}}\sum_c a_c p'_{\epsilon,c}f''_c
$$

$$
G(X,Y) = \frac{2}{(\sum_c a_c p_{\epsilon,c})^3}\sum_c a_c p_{\epsilon,c}f'_c \sum_c a_c p'_{\epsilon,c}\sum_c a_c p'_{\epsilon,c}f'_c
$$

$$
+ \frac{1}{(\sum_c a_c p_{\epsilon,c})^2}\left(-2\sum_c a_c p''_{\epsilon,c}f'_c\sum_c a_c p'_{\epsilon,c}f'_c - \sum_c a_c p'_{\epsilon,c}\sum_c a_c p''_{\epsilon,c}(f'_c)^2 + \sum_c a_c p'_{\epsilon,c}\sum_c a_c p'_{\epsilon,c}f''_c\right)
$$

$$
+ \frac{1}{\sum_c a_c p_{\epsilon,c}}\sum_c a_c p'''_{\epsilon,c}(f'_c)^2 + \frac{1}{\sum_c a_c p_{\epsilon,c}}\sum_c a_c p''_{\epsilon,c}f''_c
$$

and

$$
H(X,Y) = \frac{-2}{(\sum_c a_c p_{\epsilon,c})^3}\left(\sum_c a_c p'_{\epsilon,c} f'_c\right)^3 + \frac{-2}{(\sum_c a_c p_{\epsilon,c})^2}(\sum_c a_c p'_{\epsilon,c} f'_c)\sum_c a_c(p''_{\epsilon,c}(-f'_c)f_c + p'_{\epsilon,c}f''_c)
$$

$$
+ \frac{-1}{(\sum_c a_c p_{\epsilon,c})^2}\sum_c a_c p'_{\epsilon,c}f'_c \sum_c a_c p''_{\epsilon,c}(f'_c)^2 + \frac{-1}{\sum_c a_c p_{\epsilon,c}}\sum_c a_c p'''_{\epsilon,c}(f_c)^3 + \frac{2}{\sum_c a_c p_{\epsilon,c}}\sum_c a_c p''_{\epsilon,c} f'_c f''_c
$$

$$
+ \frac{-1}{(\sum_c a_c p_{\epsilon,c})^2}\sum_c a_c p'_{\epsilon,c}f'_c \sum_c a_c p'_{\epsilon,c}f''_c + \frac{1}{\sum_c a_c p_{\epsilon,c}}\sum_c a_c p''_{\epsilon,c} f'_c f''_c + \frac{-1}{\sum_c a_c p_{\epsilon,c}}\sum_c a_c p'_{\epsilon,c} f'''_c
$$

Since

$$
\frac{\partial^2 \pi/\partial X \partial Y}{\partial^2 \pi/\partial X^2} = 0
$$

We have

$$
u\frac{\partial v}{\partial X} - \frac{\partial u}{\partial X}v = 0
$$

$$
U(X,Y)(H(X,Y) + \xi''') - G(X,Y)(V(X,Y) + \xi'') = 0
$$

Thus, we have

$$
\xi''' - \frac{G(X,Y)}{H(X,Y)}\xi'' = \frac{G(X,Y)V(X,Y)}{U(X,Y)} - H(X,Y) \tag{2}
$$

$\square$

# B  Derivation of (10)

The objective function $\mathcal{J}$ reads

$$
\mathcal{J} = -\mathcal{L}(\mathbf{\Theta}|\mathbf{X},\mathbf{Y},\mathbf{\Omega}) + \lambda \log \mathrm{HSIC}_b(\mathbf{X},\mathbf{\Theta}). \tag{3}
$$

Then the gradient of $\mathcal{J}$ with respect to (w.r.t.) latent points $\mathbf{\Theta}$ can be computed as

$$\frac{\partial \mathcal{J}}{\partial [\mathbf{\Theta}]_{ij}} = \mathrm{tr}\left[\left(\frac{\partial \mathcal{J}}{\partial \mathbf{K}_{\Theta}}\right)^T \frac{\partial \mathbf{K}_{\Theta}}{\partial [\mathbf{\Theta}]_{ij}}\right], \tag{4}$$

where $\mathbf{K}_{\Theta}$ is the kernel matrix of latent points in $\mathbf{\Theta}$. $\frac{\partial \mathcal{J}}{\partial \mathbf{K}_{\Theta}}$ can be obtained by

$$\frac{\partial \mathcal{J}}{\partial \mathbf{K}_{\Theta}} = \frac{\partial - \mathcal{L}}{\partial \mathbf{K}_{\Theta}} + \frac{\partial}{\partial \mathbf{K}_{\Theta}} \lambda \log \mathrm{HSIC}_{\mathrm{b}}(\mathbf{X}, \mathbf{\Theta}). \tag{5}$$

The first term is computed as

$$-\frac{\partial \mathcal{L}}{\partial \mathbf{K}_{\Theta}} = -\frac{\partial \mathcal{L}}{\partial [\mathbf{K}_{\Theta}]_{ij}} = -\mathrm{tr}\left[\left(\frac{\partial \mathcal{L}}{\partial \tilde{\mathbf{K}}}\right)^T \frac{\partial \tilde{\mathbf{K}}}{\partial [\mathbf{K}_{\Theta}]_{ij}}\right]$$

$$= -\mathrm{tr}\left[\left(\tilde{\mathbf{K}}^{-1}\mathbf{Y}\mathbf{Y}^T\tilde{\mathbf{K}}^{-1} - D\tilde{\mathbf{K}}^{-1}\right)^T \left(\frac{\partial}{\partial [\mathbf{K}_{\Theta}]_{ij}}(\mathbf{K}_X \circ \mathbf{K}_{\Theta})\right)\right]$$

$$= -\mathrm{tr}\left[\left(\tilde{\mathbf{K}}^{-1}\mathbf{Y}\mathbf{Y}^T\tilde{\mathbf{K}}^{-1} - D\tilde{\mathbf{K}}^{-1}\right)^T \left(\frac{\partial \mathbf{K}_X}{\partial [\mathbf{K}_{\Theta}]_{ij}} \circ \mathbf{K}_{\Theta} + \mathbf{K}_X \circ \frac{\partial \mathbf{K}_{\Theta}}{\partial [\mathbf{K}_{\Theta}]_{ij}}\right)\right]$$

$$= -\mathrm{tr}\left[\left(\tilde{\mathbf{K}}^{-1}\mathbf{Y}\mathbf{Y}^T\tilde{\mathbf{K}}^{-1} - D\tilde{\mathbf{K}}^{-1}\right)^T \left(\mathbf{K}_X \circ \frac{\partial \mathbf{K}_{\Theta}}{\partial [\mathbf{K}_{\Theta}]_{ij}}\right)\right]$$

$$= -\mathrm{tr}\left[\left(\tilde{\mathbf{K}}^{-1}\mathbf{Y}\mathbf{Y}^T\tilde{\mathbf{K}}^{-1} - D\tilde{\mathbf{K}}^{-1}\right)^T \left(\mathbf{K}_X \circ \mathbf{J}^{ij}\right)\right], \tag{6}$$

where $\circ$ denotes the Hadamard product and $\mathbf{J}^{ij}$ is the single-entry matrix, 1 at $(i, j)$ and 0 elsewhere. The second term in Eq.(5) can be computed as

$$\frac{\partial}{\partial \mathbf{K}_{\Theta}} \lambda \log \mathrm{HSIC}_{\mathrm{b}}(\mathbf{X}, \mathbf{\Theta}) = \frac{\partial}{\partial \mathbf{K}_{\Theta}} \lambda \log \mathrm{tr}\left(\mathbf{K}_X \mathbf{H}\mathbf{K}_{\Theta}\mathbf{H}\right) = \lambda \frac{1}{\mathrm{tr}\left(\mathbf{K}_X \mathbf{H}\mathbf{K}_{\Theta}\mathbf{H}\right)} \mathbf{H}\mathbf{K}_X\mathbf{H}, \tag{7}$$

where $\mathbf{H} = \mathbf{I} - \frac{1}{m}\vec{1}\vec{1}^T$ and $\vec{1}$ is a $m \times 1$ vector of ones. To this stage, we have found $\frac{\partial \mathcal{J}}{\partial \mathbf{K}_{\Theta}}$ in Eq.(4).

## C  Adjusted Rand Index

This section contains the definition of adjusted rand index (ARI) [1] for reference.[2]

The ARI is the corrected-for-chance version of the Rand index [3]. Though the Rand Index may only yield a value between 0 and +1, the ARI can yield negative values if the index is less than the expected index.

### The contingency table

Given a set $S$ of $n$ elements, and two groupings or partitions (e.g. clusterings) of these elements, namely $X = \{X_1, X_2, \ldots, X_r\}$ and $Y = \{Y_1, Y_2, \ldots, Y_s\}$, the overlap between $X$ and $Y$ can be summarized in a contingency table $[n_{ij}]$ where each entry $n_{ij}$ denotes the number of objects in common between $X_i$ and $Y_j$: $n_{ij} = |X_i \cap Y_j|$.

### Definition

The adjusted form of the Rand Index, the ARI is

$$ARI = \frac{\sum_{ij}\binom{n_{ij}}{2} - \left[\sum_i \binom{a_i}{2} \sum_j \binom{b_j}{2}\right] / \binom{n}{2}}{\frac{1}{2}\left[\sum_i \binom{a_i}{2} + \sum_j \binom{b_j}{2}\right] - \left[\sum_i \binom{a_i}{2} \sum_j \binom{b_j}{2}\right] / \binom{n}{2}} \tag{8}$$

where $n_{ij}$, $a_i$, $b_j$ are values from the contingency table.

[1] Hubert, L., & Arabie, P. (1985). Comparing partitions. Journal of classification, 2(1), 193-218.
[2] https://en.wikipedia.org/wiki/Rand_index
[3] Rand, W. M. (1971). Objective criteria for the evaluation of clustering methods. Journal of the American Statistical association, 66(336), 846-850.

Table 1: Contingency table

|  | $Y_1$ | $Y_2$ | $\dots$ | $Y_s$ | Sums |
|---|---|---|---|---|---|
| $X_1$ | $n_{11}$ | $n_{12}$ | $\dots$ | $n_{1s}$ | $a_1$ |
| $X_2$ | $n_{21}$ | $n_{22}$ | $\dots$ | $n_{2s}$ | $a_2$ |
| $\vdots$ | $\vdots$ | $\vdots$ | $\ddots$ | $\vdots$ | $\vdots$ |
| $X_r$ | $n_{r1}$ | $n_{r2}$ | $\dots$ | $n_{rs}$ | $a_r$ |
| Sums | $b_1$ | $b_2$ | $\dots$ | $b_s$ | |

# D   Clustering Results Visualization

In this section, clustering results with ARI of ANM-MM close to avgARI[4] shown in Table 1 are visualized. Results of comparing approaches on the same data are also given.

## D.1   Experiments different generating mechanisms and sample size

The ground truth and clustering results of all approaches in one of the 100 independent experiments are visualized in Fig. 1.

## D.2   Experiments on different number of generating mechanisms

The ground truth and clustering results of all approaches in one of the 100 independent experiments are visualized in Fig. 2.

## D.3   Experiments on different noise standard deviation

The ground truth and clustering results of all approaches in one of the 100 independent experiments are visualized in Fig. 3.

## D.4   Experiments on different mixing proportions

The ground truth and clustering results of all approaches in one of the 100 independent experiments are visualized in Fig. 4.

(a) Ground truth $f_1$    (b) $f_2$    (c) $f_3$    (d) $f_4$

(e) ANM-MM $f_1$    (f) $f_2$    (g) $f_3$    (h) $f_4$

(i) $k$-means $f_1$    (j) $f_2$    (k) $f_3$    (l) $f_4$

(m) GMM $f_1$    (n) $f_2$    (o) $f_3$    (p) $f_4$

(q) Spectral clustering $f_1$    (r) $f_2$    (s) $f_3$    (t) $f_4$

(u) DBSCAN $f_1$    (v) $f_2$    (w) $f_3$    (x) $f_4$

Figure 1: Clustering results different type of mechanisms. The first row shows the ground truth and remaining rows correspond to different clustering approaches. Each column corresponds to a generating mechanism.

(a) Ground truth 2 mechanisms

(b) 3 mechanisms

(c) 4 mechanisms

(d) ANM-MM of 2 mechanisms

(e) 3 mechanisms

(f) 4 mechanisms

(g) $k$-means of 2 mechanisms

(h) 3 mechanisms

(i) 4 mechanisms

(j) GMM of 2 mechanisms

(k) 3 mechanisms

(l) 4 mechanisms

(m) SpeClu of 2 mechanisms

(n) 3 mechanisms

(o) 4 mechanisms

(p) DBSCAN of 2 mechanisms

(q) 3 mechanisms

(r) 4 mechanisms

Figure 2: Clustering results on different number of mechanisms. The first row shows the ground truth and remaining rows correspond to different clustering approaches. Each column corresponds to a number of generating mechanisms.

(a) Ground truth $\sigma = 0.01$      (b) $\sigma = 0.05$      (c) $\sigma = 0.1$

(d) ANM-MM $\sigma = 0.01$      (e) $\sigma = 0.05$      (f) $\sigma = 0.1$

(g) $k$-means $\sigma = 0.01$      (h) $\sigma = 0.05$      (i) $\sigma = 0.1$

(j) GMM $\sigma = 0.01$      (k) $\sigma = 0.05$      (l) $\sigma = 0.1$

(m) SpeClu $\sigma = 0.01$      (n) $\sigma = 0.05$      (o) $\sigma = 0.1$

(p) DBSCAN $\sigma = 0.01$      (q) $\sigma = 0.05$      (r) $\sigma = 0.1$

Figure 3: Clustering results on different noise standard deviations. The first row shows the ground truth and remaining rows correspond to different clustering approaches. Each column corresponds to a value of $\sigma$.

Figure 4: Clustering results different mixing proportions. The first row shows the ground truth and remaining rows correspond to different clustering approaches. Each column corresponds to a value of $a_1$.

## Footnotes

[4]in the sense that $|\text{ARI} - \text{avgARI}| < 0.05$