[Reviews · NeurIPS 2018]

Reviewer 1



This paper generalize the additive Noise Model to a mixture model and propose a clustering mechanism to find the latent class and a GPPOM to learn the latent parameters. This paper also includes an identifiability condition for ANM-MM. The structure is clear and well-organized. However, I have some concerns that need further clarification from the authors: - In line 76, the assumption about $\theta $ independent of the cause X according to postulate 1, which may not holds, because the postulate 1 assume that the function f is independent to the DISTRIBUTION of cause but not independent the cause. - The theorem 1 is proven under an implicit assumption that we have already known the latent cluster c for each sample. However, there is no guarantee that the clusters are recoverable even when the k-means reach the global optimum. - In the experiments, this method seems unable to run on a larger samples size, which may be restrictive and unconvincing. Especially, the experiments on the Tuebingen cause-effect pairs, samples size of which are range from a hundred to ten thousand, however, this paper only samples 90 points on each pair. words: - The two abbreviations -- generating mechanism (g.m.), random variables (r.v.s) -- are not commonly used, which is not a good style. typos: - Line 59: is it $a_{c} \geq 0$? - Line 156: an sample =>a sample

Reviewer 2



This paper proposes an approach to estimate causal relationships between two variables X and Y when properties of the mechanism changes across the dataset. The authors propose and extension of the non-linear additive noise model [Hoyer et al. 2009] to the case of a mixture of a finite number of non-linear additive noise models, coined Additive Noise Model- Mixture Model (ANM-MM). The authors propose a theoretical identifiability result based on the proof of [Hoyer et al. 2009], then provide an estimation algorithm based on Gaussian Process Partially Observable Models (GPPOM), introduced as a generalization of Gaussian Process Latent Variable Models (GPLVM). Comparison of the approach to baseline for causal inference and clustering are provided on real and simulated data. The problem addressed in this paper is definitively interesting. While some of the experimental results are promising, theoretical and empirical provide a limited understanding of the approach, which is rather complex, and in particular of its strength and limitations. Main concerns follow. Theoretical results: It is unclear to me why the theoretical identifiability result of Theorem 1 is not a very simple corollary of [Hoyer et al., 2009]: assume both directions admit an ANM-MM, then there is a pair of function parameters values (\theta,\omega) corresponding to each direction, that happen simultaneously with non-zero probability. Then conditioning on this event, we get an ANM for both directions and can conclude that this cannot happen in a generic case, based on [Hoyer et al., 2009]. Am I missing something? At the same time, this theoretical result does not offer guaranties for finite samples. Given the complexity of the model, I can imagine several things that can go wrong when applying the model: - When the number of clusters gets too large, overfitting of the models (due to lack of samples) may reduce the information in additive noise that can be used for causal inference. How can we assess (theoretically or experimentally) and possibly handle this issue? This is partially shown in Fig. 3a), but the study restrict itself to the case of a number of clusters inferior or equal to ground truth. On the other hand, there are baseline methods that do not assume a mixture of mechanisms (such as LiNGAM or PNL) that seem quite robust to variations in the mechanism. - In the particular context of ANM, it would be good that the authors provide a toy example showing that mixing mechanisms can lead to misleading results. Intuitively, not capturing the variation in the mechanism with lead to an increase in additive noise. Is there any toy example where one can prove ANM would fail? Experimental results Regarding simulation, one simple baseline should use clustering combined with ANM inference (all “standard” causal inference approaches can handle a single mechanism). Moreover, clustering (such as K-means) in the PCA domain seems a fair and reasonable baseline that is likely to perform better than in the original space. Given the complexity of the algorithm, it seems difficult to choose the hyperparameters for a given dataset (this includes number of clusters and parameter of independence). How to we choose them to reproduce the results on the Tuebingen dataset? Also it is unclear whether clustering of mechanism is necessary in most of these datasets, so I wonder if it would not be worth focusing on more datasets where the mechanism variability is known. The FOEN data seems to be an interesting example, and I would suggest to put part of supplemental Fig.5 in main text.

Reviewer 3



The paper extends the additive noise model for causal inference, which assumes independece of mechanism and noise but allows for nonlinear fucntional forms and non-Gaussian noise, to the case where the data are generated from multiple causal mechanisms with the same causal relations but different fuctional parameters, ANM-MM. The paper provides a type of of identifiability condition more restrictive than the general ANM and propopses a procedure for fitting the mixture model. The approach appears to be sound and reasonable given theoretical results from previous approaches. The empirical results are generally good, though not distinguishable from non-mixture approachs, e.g. ECP, in all simulations. The application to the cause-effect pairs dataset without significant discussion is confusing, however - are each of these paris assumed to be generated from multiple models? How are the number of models determined? There is a worry that by adding different numbers of mixtures without explicit motivation this is overfitting the dataset. In general the paper is well written and the procedure is well explained. The paper could do a better job motivating the approach earlier on. We dont get a clear motivating example until the end of the paper, the BAFU air data. Prior to this, ti was not clear whether the paper was attempting to force a combination of ANMs with GP-LVM without clear motivation. Summary: The paper extends the additive noise model to the case where data are obtained from multiple casual models with the same causal relations but different functional parameters. The paper is sound, well written, and novel, though it's not clear how much this advances the state of the art given questions regarding how much more accurate it is than non-mixture approaches and how widely applicable this approach is.